# Corrosion of Steel Rebars in Anoxic Environments. Part II: Pit Growth Rate and Mechanical Strength

**DOI:** 10.3390/ma14102547

**Published:** 2021-05-14

**Authors:** Elena Garcia, Julio Torres, Nuria Rebolledo, Raul Arrabal, Javier Sanchez

**Affiliations:** 1Eduardo Torroja of Construction Science Institute (IETcc-CSIC), Serrano Galvache, 4, 28033 Madrid, Spain; elenacampaspero@gmail.com (E.G.); juliotorres@ietcc.csic.es (J.T.); nuriare@ietcc.csic.es (N.R.); 2Department of Chemical Engineering and Materials, Faculty of Chemistry, Complutense University of Madrid, Plaza de las Ciencias, 28040 Madrid, Spain; rarrabal@ucm.es

**Keywords:** corrosion, anoxic conditions, reinforced concrete, chloride, pitting

## Abstract

Reinforced concrete may corrode in anoxic environments such as offshore structures. Under such conditions the reinforcement fails to passivate completely, irrespective of chloride content, and the corrosion taking place locally induces the growth of discrete pits. This study characterised such pits and simulated their growth from experimentally determined electrochemical parameters. Pit morphology was assessed with an optical profilometer. A finite element model was developed to simulate pit growth based on electrochemical parameters for different cathode areas. The model was able to predict long-term pit growth by deformed geometry set up. Simulations showed that pit growth-related corrosion tends to maximise as cathode area declines, which lower the pitting factor. The mechanical strength developed by the passive and prestressed rebar throughout its service life was also estimated. Passive rebar strength may drop by nearly 20% over 100 years, whilst in the presence of cracking from the base of the pit steel strength may decline by over 40%.

## 1. Introduction

Offshore, port and deep geological nuclear waste storage structures must be built with reinforced concrete designed for long-term durability in oxygen-free environments. Part 1 of this study addressed the electrochemistry of reinforcing steel embedded in concrete located in an anoxic environment [1,2,3]. Corrosion rate was observed to be governed and limited by the cathode reaction and unaffected by the chloride concentration in the concrete at concentrations from 0 wt.% to 2 wt.%. The Tafel slopes and limiting current density were calculated and corrosion was found to concentrate in discrete pits.

This second part deals with the geometric characterisation of such experimentally generated pits and the simulation of their growth by entering experimentally determined electrochemical values into a finite elements model (FEM) [4,5,6,7,8]. Model validation provided grounds for the long-term extrapolation of pit development. 

Earlier findings were also used to estimate the variation in mechanical strength induced by pitting or related cracking [7,8]. Pit growth and geometry were then used to determine steel ultimate tensile strength and draw a damage tolerance diagram factoring in the effect of associated developments such as hydrogen embrittlement [6,9,10,11,12,13].

## 2. Materials and Methods

### 2.1. Materials

The six 10 × 10 × 10 cm^3^ cubic concrete specimens used for this study were prepared with 350 kg of type I 42.5 R cement per m^3^ of concrete. Aggregate batching per m^3^ of concrete was: fines (0 mm to 4 mm), 766.00 kg/m^3^; coarse aggregate (4 mm to 12 mm), 823.6 kg/m^3^; and gravel (12 mm to 20 mm), 325.60 kg/m^3^, whilst the water/cement ratio was 0.45. The mixing water used contained chlorides in the form of sodium chloride (NaCl) at concentrations ranging from 0% to 2%, referred to cement weight. The specimens were prepared by pairs, with specimens E01 and E02 bearing no chlorides; E11 and E12 1 wt.% Cl^−^ and E21 and E22 2 wt.% Cl^−^. All specimens bore an embedded 6 mm diameter B500SD steel bar. After curing for 1 d in a humidity chamber they were removed from the moulds and submerged in a 30 g/L solution of NaCl and stored in a glovebox through which nitrogen was flowed to simulate an anoxic environment.

### 2.2. Methods

Upon conclusion of the 232 d test, the electrochemical values (see Part 1) were recorded and the specimens were oven-dried and broken open. The steel surface was immediately analysed under an optical microscope to identify the pits generated. The rebar was subsequently pickled to remove the oxide from the pits. Pit morphology was assessed with an Alicona InfiniteFocusSL (Bruker) focus variation optical profilometer fitted with a 10× magnifying lens and featuring vertical resolution at 0.1 µm. A profilometer scan of the pits previously identified and labelled delivered information on the form and depth of the pits.

The findings described here and in part 1 of this article were applied to develop a finite element model to simulate pit growth based on electrochemical parameters for different cathode areas. The model was able to predict long-term pit growth based on calculations performed with COMSOL Multiphysics 6.5 software [14,15,16]. The ‘secondary current distribution’ interface used was compatible with both Ohm’s law and the Tafel or Butler–Volmer equations for calculating electrode charge transfer.
(1)$$\nabla -\rho^{-1}\left(\nabla \emptyset \right)=Q$$
where *ρ* is the resistivity, *φ* is the potential field and *Q* is the external current.

Current density, *i*, follows: (2)$$i=-\rho^{-1}\left(\nabla \emptyset \right)$$

The activation overpotential, denoted *η*, is the following:(3)$$\eta =\emptyset -E_{eq}$$
where *E_eq_* denotes the equilibrium potential.

The anodic and cathodic Tafel equation is implemented as follows:(4)$$i=-i_{0}\;10^{\eta /A}$$
where $i_{0}\;$ is exchange current density and *A* is the Tafel slope.

The model also deployed ‘deformed geometry’ from which pit growth could be simulated from the values obtained in the preceding modulus. Inasmuch as corrosion adopted the form of pits in this study, which aimed to predict their growth, 2D-axisymmetric (i.e., simulated 3D with symmetrical rotation) geometry was used. The FEM simulation scheme depicted in Figure 1 shows the medium, concrete in this case, the anode and cathode areas and the axis of symmetry or rotation.

The electrochemical parameters used in the model, determined in part 1, are summarised in Table 1. On the one hand, the medium, which in this case is concrete, is taken into account through the value of its resistivity (*ρ*) which was measured in the experiments. The cathodic reaction is defined by the cathode equilibrium potential (*E_eq,c_*), cathode exchange current density (*i*_0,*c*_) and the cathode Tafel slope (*A_c_*). Likewise, the anodic reaction has been defined through the anode equilibrium potential (*E_eq,a_*), anode exchange current density (*i*_0,*a*_), anode Tafel slope (*A_a_*) and anode limiting current density (*i_lim,a_*).

The MUMPS (multifrontal massively parallel sparse direct solver) solver was applied, defining step times of under 1 d over 100 years. Intermediate findings were automatically remeshed as the pit grew. The mesh used was a triangular element defining the cathode area with at least 100 elements. The full mesh consisted in over 500 elements, generating around 8000 degrees of freedom, as shown in Figure 2. The convergence tolerance criterion was set at 10^−3^.

Calculations were performed at a work station featuring two Intel Xeon E5-2690 v3 12-core CPU processors and a 125 GB RAM.

Calculations were run for different anode area/cathode area ratios at a fixed anode area radius of 50 µm and varying the cathode area from 0.1 cm to 10 cm.

## 3. Results and Discussion

### 3.1. Pit Geometry

Irrespective of the chloride content in the specimen, optical microscopy revealed a mean of three pits on each bar, subsequently assessed with a profilometer. In the scans reproduced in Figure 3 and Figure 4 by way of example, the rebar is depicted both in its natural colour on the left and colour-coded to denote depth on the right. In one case, sample E12, pitting has been generated next to the rebar ribs, while in another case, sample E21, pitting has been generated between the ribs. Two examples have been taken to show that pitting growth can occur anywhere on the rebar.

Pit geometry was analysed with the profilometer findings. From the 3D geometry of the pit, the profile was obtained by cutting through a plane containing the pit growth axis. Once this profile had been obtained, the pit was fitted to an ellipsoid defined by two perpendicular axes, one that defines the depth of the pit and the other that represents the lateral growth of the pit. Figure 5 shows two examples, in which pit growth was defined along two perpendicular axes, *Py* (half-width) and *Px* (pit depth).

Table 2 lists the values of the two parameters, depth (*Px*) and half-width (*Py*), in µm, for all the pits found on the bars embedded in concrete specimens. Since the differences found among them were not significant, all the pits were deemed to constitute a single population. The mean 232 d depth of the pits identified was found to be 97 µm. That information is of immense utility in long-term structural durability studies, for it provides insight into rebar degeneration that can be used as grounds for establishing its service life.

### 3.2. Finite Element Simulation of Pit Growth

The findings delivered by COMSOL Multiphysics finite element software are discussed in this section. Figure 6 shows pit growth, the current lines between anode and cathode and their distribution in the concrete for a 2 mm cathode area with time = 360 d. The figure also shows the variation in pit dimensions between 360 d and 720 d. Pit growth generated an ellipsoid, one of whose axes being depth and the other two half-width.

Figure 7 graphs the variation in pit depth with time as simulated with FEM for all the simulated cathode areas. Since the cathode reaction governed the process, for any given time pit depth increased with the cathode area, as expected. At the end of the 232 d experiment the cathode area was observed to have a 1 mm to 2 mm radius.

As the experimental data show, the pits were not spherical but ellipsoid. Growth was likewise ellipsoid according to the FEM findings. The findings for a pit with a 1 mm cathode radius are plotted in Figure 8. Pit half-width, *Py*, was observed to be greater than depth, *Px*. The *Py/Px* ratio calculated from the experimental data was 1.45.

The *Px* and *Py* values found with the FEM simulations were used to calculate pit volume under the assumption that the pits were spheroids or, equivalently, ellipsoids of revolution, whereby:(5)$$V_{corr}\left(t\right)=\frac{4}{3}\pi \;Px\left(t\right)\;{\left(Py\left(t\right)\right)}^{2}$$

The simulated data were applied to calculate corrosion rate (*i_corr_*) using Faraday’s law [2]:(6)$$i_{corr}\left(t\right)=\frac{F\;z\;\rho }{A\;P}\;\frac{\partial V_{corr}\left(t\right)}{\partial t}$$
where *F* is the Faraday constant, *z* the number of electrons involved in the anode reaction, *ρ* density, *A* area, *P* molecular weight, *V* pit volume and *t* time.

The variation in corrosion rate over time for the simulated cathode areas is shown below. Although as a rule the lower the cathode area the higher the rate, due to pit growth *i_corr_* peaked at a 2 mm cathode radius and an age of around 40 years.

For that reason, whilst greater pit depth was favoured by greater cathode area (Figure 7), as Figure 9 shows, the peak corrosion rate was reached at the minimum cathode area. Further to the experimental data, the predominant parameter was corrosion rate; i.e., in anoxic corrosion the rate maximised while cathode area tended to minimise. This study showed that the cathode area was defined by a circle with a 1 mm to 2 mm radius, inferring that pit growth took place every 3.1 mm^2^ to 12.6 mm^2^. 

Pitting factor *α* is defined as the ratio between mean pit depth and mean thickness loss due to pervasive corrosion [4]. That parameter is plotted across a full 100 year service life for all the cathode radii studied in Figure 10. Very high factor values were observed at the outset of corrosion and when the cathode area was greatest. For a 1 mm or 2 mm cathode area, the *α* values were under 10 and tended to decline to 1 to 4 in the longer term.

### 3.3. Rebar Mechanical Performance

Earlier reports [7,8,17,18,19] and the present findings for anoxic conditions were used to estimate the mechanical performance of concrete-embedded steel rebar. Further to finite element model predictions, growth was expected in a pit with an aspect ratio *P_y_/P_x_* of approximately 1.2 (Figure 11), a value consistent with the experimental data for long periods of time.

The forecasts for up to 100 years’ pit growth for the cathode areas studied are graphed in Figure 12. In keeping with the experimental results, only the mechanical performance estimates for cathode area radii of 0.1 cm to 0.2 cm are discussed below. 

Two scenarios were envisaged to estimate mechanical performance: growth of a pit with an aspect ratio of 1.2; and cracking initiated at the pit. 

Variation in mechanical strength was calculated for the following conditions and assumptions: rebar diameter, 6 mm; steel fracture toughness, 50 MPa·m^1/2^ [7,20,21]; cold-drawn pearlitic steel (commonly used in prestressed structures) ultimate strength, 1800 MPa [19,22,23].

Where induced by pit growth, ductile failure is found with the following equation:(7)$$F_{u}/F_{u0}=1.0-0.0636\left(\frac{P_{x}}{R}\right)-0.0065\left(\frac{P_{y}}{R}\right)-1.6050{\left(\frac{P_{x}}{R}\right)}^{2}+0.0004{\left(\frac{P_{y}}{R}\right)}^{2}$$
where *F_u_*/*F_u_*_0_ is the ratio between ultimate strength in the presence of a pit and ultimate strength in a non-deteriorated rebar, and *R* is rebar radius.

For steel bars, the stress intensity factor (*K_I_*) and consequently the failure criterion can be calculated with the Valiente and Elices equation [1,16,17]. Assuming the presence of semi-ellipsoid surface cracks, those authors developed the following expression to determine the stress intensity factor with the principal semi-axes *Px* (depth) and *Py* (half-width):(8)$$\frac{K_{I}}{\sigma \sqrt{\pi P_{X}}}=\overset{4}{\underset{i=0\;i\neq 1}{\sum }}\overset{3}{\underset{j=0}{\sum }}C_{ij}{\left(\frac{P_{X}}{d}\right)}^{i}{\left(\frac{P_{X}}{P_{y}}\right)}^{j}$$
where *α* is stress, *C_ij_* are constants and *d* is diameter.

The failure diagram in Figure 13 shows the decline in steel ultimate strength for cathode radii of 0.1 cm and 0.2 cm. Two scenarios were envisaged: steel ductile failure due to pit growth and brittle failure due to cracking at the base of the pit, assuming the crack to be the same size as pit [5,9,10]. Pit growth-induced ductile failure was deemed to entail plasticisation of the remaining section. In that case failure would take place at a stress level of 80% to 90% of the ultimate strength in the 100th year of service life. In the second scenario, with a crack emanating from the pit, failure mechanics had to be taken into consideration and the stress intensity factor for surface cracks in a cylindrical geometry calculated [8,18,24,25]. The hydrogen effect was also deemed to be present, lowering failure toughness [7,13,26,27,28] due to the weakening of inter-iron bonds [9,10,11,12].

The calculations performed under those conditions predicted that failure would be reached at a load of 60% to 70% of the ultimate strength in the 100th year of service life. The significant decline in strength in that case could obviously lead to sudden failure, which would entail greater risk for members exposed to high stress.

## 4. Conclusions

The conclusions to be drawn from rebar corrosion geometry in anoxic environments are set out below.

A mean of three pits per bar was found, irrespective of chloride content. Pit morphology fitted an ellipsoid pattern. The mean pit depth in 232 d specimens was 97 µm.FEM simulation showed that pit growth-related corrosion tends to maximise as cathode area declines, although the largest pits did not form under those conditions, which lower the pitting factor.Simulation afforded estimates of rebar mechanical performance across a structure’s service life. In a passive rebar, strength may decline by nearly 20% over 100 years, whilst in the presence of cracking from the base of the pit, which may occur in an active (prestressed) rebar, steel strength may decline by over 40%.

## Figures and Tables

**Figure 1 materials-14-02547-f001:**
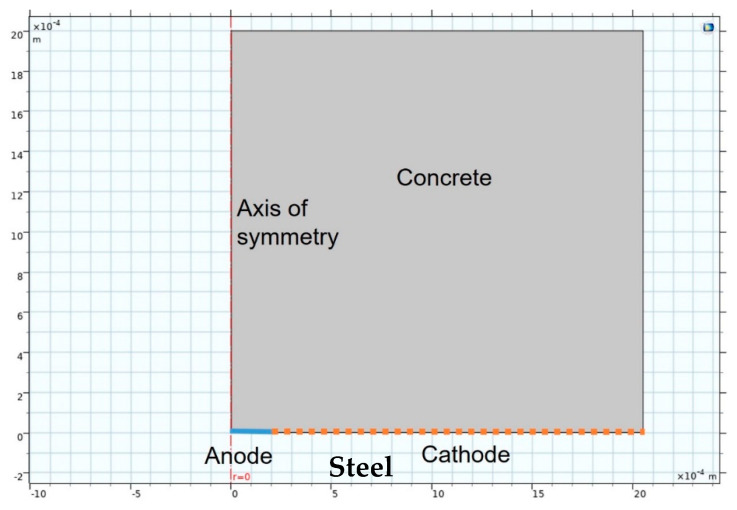
Finite elements model (FEM) simulation geometry.

**Figure 2 materials-14-02547-f002:**
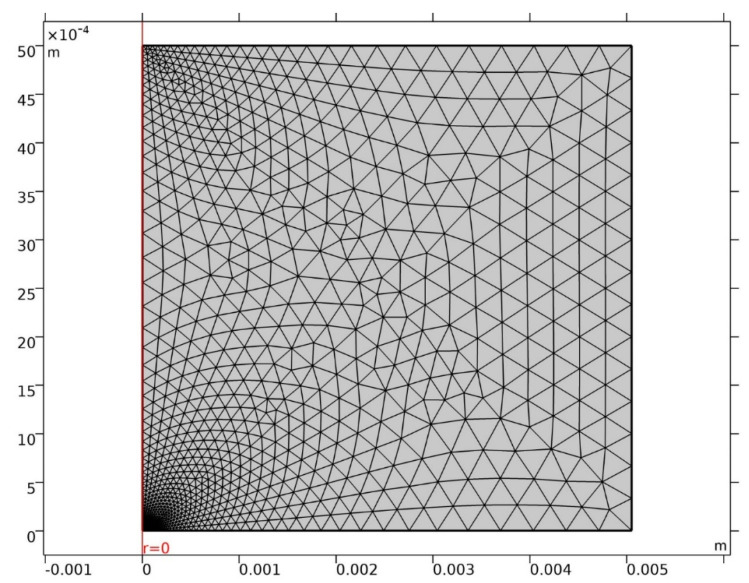
Mesh used.

**Figure 3 materials-14-02547-f003:**
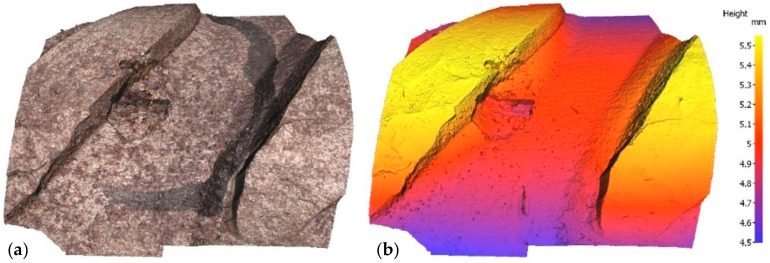
Pit in specimen E12: (**a**) natural colour; (**b**) depth-based colour coding.

**Figure 4 materials-14-02547-f004:**
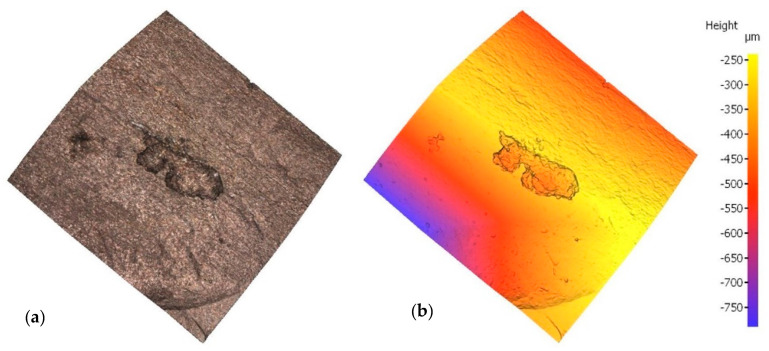
Pit in specimen E21: (**a**) natural colour; (**b**) depth-based colour coding.

**Figure 5 materials-14-02547-f005:**
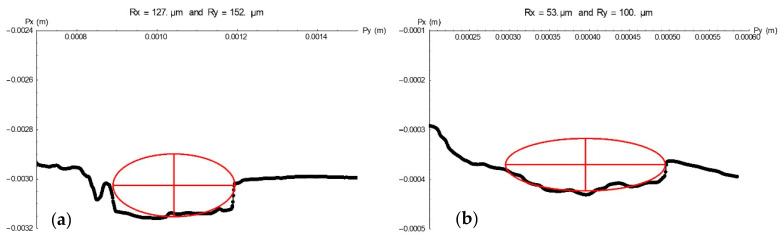
Pit profiles for specimen E12 (**a**) and E21 (**b**) as determined by plotting *Py* (pit half-width) against *Px* (pit depth) values.

**Figure 6 materials-14-02547-f006:**
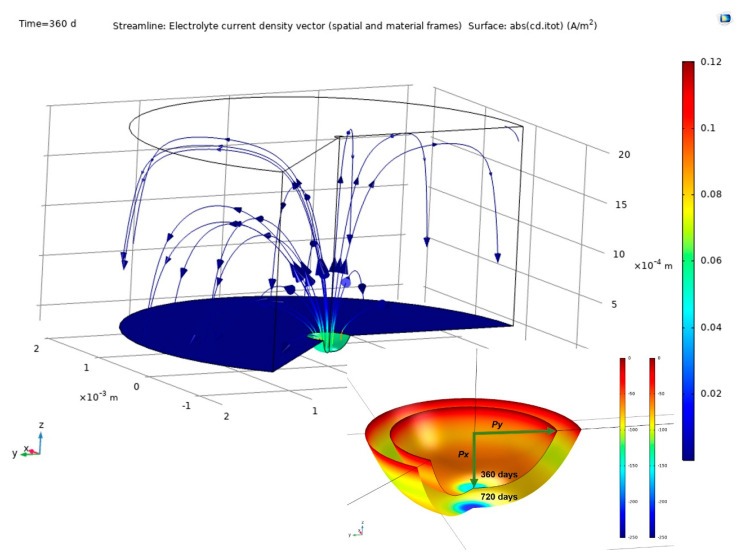
Above, simulation findings: pit growth and current lines at the electrodes and through the concrete (2 mm cathode radius, time = 360 d); below, variation in pit from 360 d to 720 d.

**Figure 7 materials-14-02547-f007:**
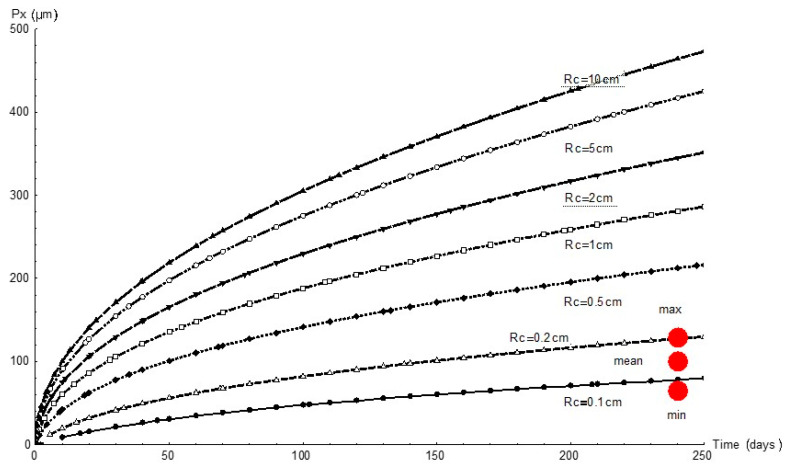
Variation in pit depth, *Px*, over time for different cathode areas and radii, *Rc*, and comparison with experimental values (red circles).

**Figure 8 materials-14-02547-f008:**
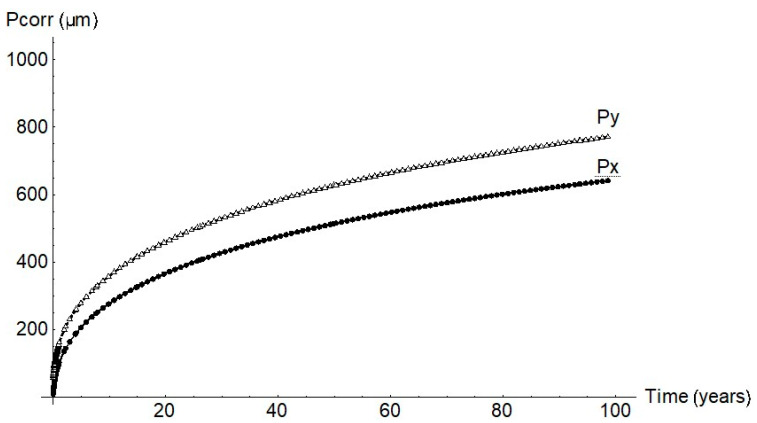
Pit geometry for a cathode radius of 1 mm.

**Figure 9 materials-14-02547-f009:**
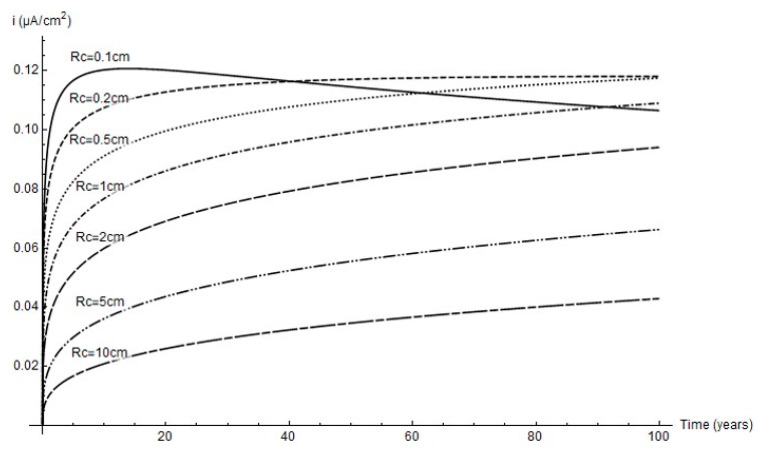
Variation in corrosion rate over time at different cathode radii.

**Figure 10 materials-14-02547-f010:**
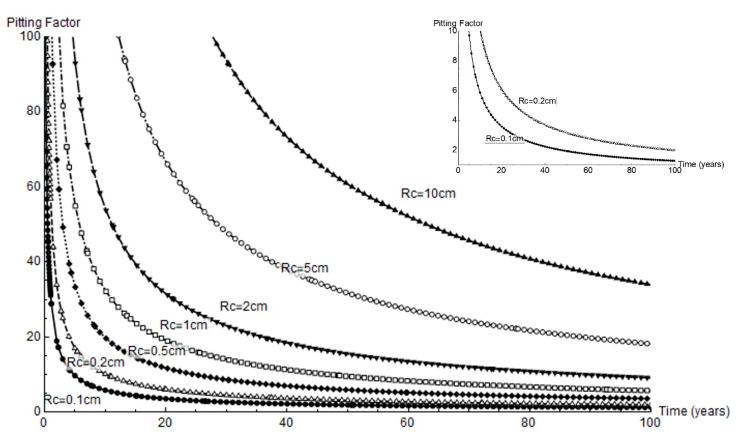
Variation in pitting factor over time for the geometries studied; right, variation for a cathode area with radius = 0.1 cm to 0.2 cm.

**Figure 11 materials-14-02547-f011:**
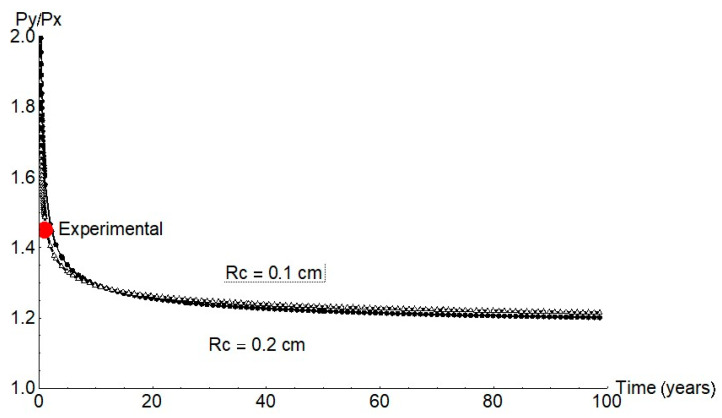
*P_y_*/*P_x_* ratio for 1 mm and 2 mm cathode radii (red circle = mean experimental value).

**Figure 12 materials-14-02547-f012:**
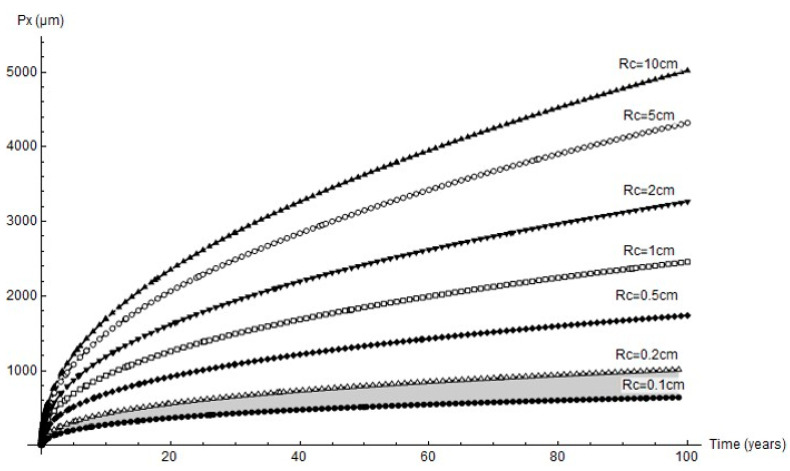
Predicted variation in pit depth for different cathode areas over long time periods (grey shading = range of experimental findings).

**Figure 13 materials-14-02547-f013:**
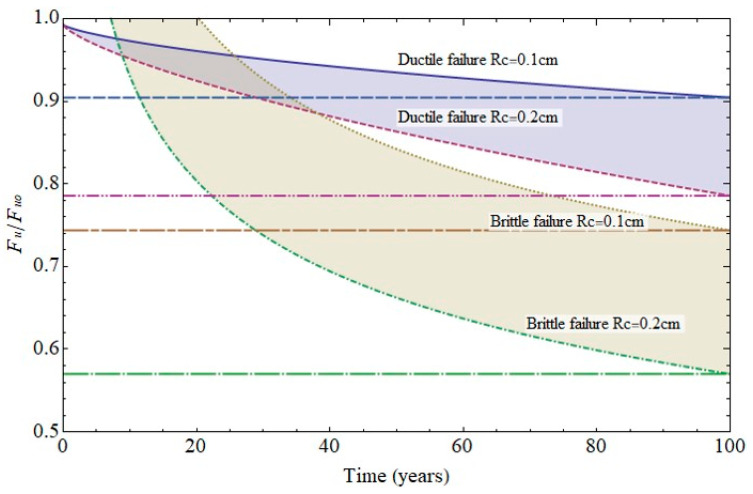
Variation in (6 mm diameter) rebar strength over time due to pitting (ductile failure) or cracking (brittle failure).

**Table 1 materials-14-02547-t001:** Summary of Tafel slopes and current densities found in the literature.

Parameter	Value	Description
*ρ*	185 Ω m	Concrete resistivity
*E_eq,c_*	−197 mV_Ag/AgCl_	Cathode equilibrium potential
*i* _0,*c*_	2.67 × 10^−5^ A/m^2^	Cathode exchange current density
*A_c_*	−185 mV	Cathode Tafel slope
*E_eq,a_*	−637 mV_Ag/AgCl_	Anode equilibrium potential
*i* _0,*a*_	3.75 × 10^−4^ A/m^2^	Anode exchange current density
*A_a_*	60 mV	Anode Tafel slope
*i_lim,a_*	0.98 × 10^−2^ A/m^2^	Anode limiting current density

**Table 2 materials-14-02547-t002:** Rebar pit depth (*Px*) and half-width (*Py*) (in µm).

No.	*Px*/*Py*					
	E01	E02	E11	E12	E21	E22
**1**	50.5/59.3	78/80	96/170	103/170	174/197	127/152
**2**	60/54	76/90	81.5/69.2	155/313	132/180	50/76
**3**	---	51/75	101.6/183	127/152	53/100	126/163
**4**	---	---	---	118/245	63/120	124/150

## Data Availability

Data available in a publicly accessible repository.

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
