# Peer review of "Corrosion of Steel Rebars in Anoxic Environments. Part II: Pit Growth Rate and Mechanical Strength"

_materials, 2021, doi:10.3390/ma14102547_

Round 1
Reviewer 1 Report
In presented manuscript the authors analysed a reinforced concrete may corrode in anoxic environments. They stated that the under such conditions the reinforcement fails to passivate completely, irrespective of chloride content, and the corrosion taking place locally induces the growth of pits. In this manuscript the authors characterised such pits and simulated their growth from empirically determined electrochemical parameters. Also the mechanical strength developed by the rebar throughout its service life was also estimated.
The presented manuscript of the authors are very interesting and knowledge about it can help in the selection of operational parameters in a specific corrosive environment. The some issue are not clearly described in manuscript. Therefore before publishing should be consider the following comments:
- In my opinion the abstract is too general. Please enrich it with the obtained data.
- Please expand the introduction with 2-3 publications referring to the results presented in the manuscript.They may be publications of other authors or their own.
- It will be more readable if will be presents parameters are summarized in the table.
- Figures 1 and 2 are too large.
- Figures 3, 4 and 5 require more comment.
- Can the structure with steel rebars i still be used after 100 years exploatation? Are in the opinion of the authors the effects of corrosion are destructive? As is known, chlorine is an aggressive medium. Please for comment on this issue.
Reviewer 2 Report
The manuscript entitled “Corrosion of steel rebars in anoxic environments. Part II: pit growth rate and mechanical strength” characterized the growth of discrete pits induced in steel bars due to corrosion and simulated their growth from empirically determined electrochemical parameters.
The simulation lacks clarity, and more details should be provided. Moreover, the presentation should be improved based on the following technical comments.
Technical Comments:
- The manuscript could benefit greatly from professional editing to improve technical writing and English.
- The literature review is poor. I understand that this manuscript is the second part of a previous version. However, the authors should increase their discussion on previous related research and highlight how their study is providing a different approach or adding significantly to what has been done.
- Half of the abstract is an introduction about the corrosion of steel reinforcement. It does not make sense. The abstract should be re-phrased to include the methodology used in this investigation and to highlight the most significant conclusion.
- Line 36: What do you mean by 10 cm3? Is it the volume of the specimen?
- Line 38: This dosage of the fine aggregate does not match with that on in the manuscript part 1. Why?
- Line 39: What was the water/cement ratio? This information should be provided with concrete ingredients.
- Line 43: "6 mm" is the diameter of the steel bar?
- Section 2.2: The authors presented most of the information about the geometry of the used FE model. However, the constitutive models of used materials and modeling of the contact between the steel bar and concrete surfaces should be highlighted.
- Line 61: "for calculating electrode transfer charge transfer" Does it need correction?
- Figure 1: Can I ask where is the steel bar in the shown scheme?
- Section 2.2: All elements in this model were made from the same type? However, there are concrete and steel bar.
- Line 80: Could you highlight what is the DOF for each node of the used element? The authors mentioned 8000 DOF as a total number.
- This is the first time for this reviewer to see the right and left directions to discuss a figure. I think it will be better to label your figures as a, b, c, ....
- What do you mean by empirical values? Do you mean values from experimental? The authors should clarify this.
- Line 159: A reference should be cited for Faraday’s law.
- In the conclusion section, please clearly list the new key findings supported by the numerical investigation.
Round 2
Reviewer 1 Report
The authors presented answers to my comments. I recommend this manuscript to publish.
Reviewer 2 Report
The authors have addressed most of the comments highlighted by the reviewer except the last comment related to the conclusion section. However, the manuscript is fine and can be accepted for publication in this journal.